# Mechanical Properties and Toughening Mechanisms of Promising Zr-Y-Ta-O Composite Ceramics

**Xiaoteng Fu** [1,2], **Fan Zhang** [1,2], **Wang Zhu** [1,2] and **Zhipeng Pi** [1,2,*]

1   Key Laboratory of Low Dimensional Materials Application Technology, Xiangtan University, Ministry of Education, Xiangtan 411105, China; 202021001597@smail.xtu.edu.cn (X.F.); zhangfan15@xtu.edu.cn (F.Z.); wzhu@xtu.edu.cn (W.Z.)
2   School of Materials Science and Engineering, Xiangtan University, Xiangtan 411105, China
*   Correspondence: pizhipengmath@163.com

**Abstract:** $ZrO_2$-$YO_{1.5}$-$TaO_{2.5}$ (ZYTO) composite ceramics are considered to be a candidate for next-generation thermal barrier coatings (TBCs) due to their excellent thermal stability and low thermal conductivity in high temperatures; however, the mechanical properties and fracture toughness of the ZYTO system may be shortcomings compared with 7-8YSZ: the traditional TBC. In this study, ZYTO composite ceramics were successfully prepared by chemical coprecipitation reaction, and the microstructure of resulting composites was studied as a function of the doping of M-YTaO$_4$. Mechanical properties, including the density, porosity, hardness and Young's modulus, were all determinate; the toughening mechanism was verified by the crack growth behavior of the Vickers indentation test. The results suggest that M-YTaO$_4$ refined the fluorite phase grain and strengthened the grain interface in the composite ceramic. The thermal mismatch between the second phase and matrix produced residual stress in the bulk and affected the crack propagation behavior. With the increase in M-YTaO$_4$ doping, the grain coarsening and ferroelastic domains were observed in the experiments. The ferroelastic domains with orthogonal polarization directions near the crack tip evidenced the ferroelastic toughening mechanism. The competition among these crack behaviors, such as crack deflection, bridging and bifurcation, dominated the actual fracture toughness of the composite. The best toughening formula was determined in the two-phase region, and the highest indentation fracture toughness was about 42 J/m$^2$, which was very close to 7-8YSZ's 45 ± 5 J/m$^2$.

**Keywords:** mechanical properties; microstructure; toughness; crack propagation

## 1. Introduction

TBC has been widely used on engine hot-end components in order to improve the efficiency and thrust-to-weight ratio of aero-turbine engines [1]. It is a system of multi-layer materials that are combined to protect engine hot-end components from high-temperature environments [2]. For TBC materials, a lot of research has been published, mainly focusing on thermophysical and mechanical properties. Low thermal conductivity and high-temperature thermostability can improve engine thermal efficiency. Excellent mechanical properties can improve the durability and wear resistance of TBCs [3].

Many researchers have conducted a lot of research in the field of TBC composites. The c-$ZrO_2$ and $La_2Ce_2O_7$ solid solution with fluorite-type structure powders were mixed and hot-pressed pyrochlore phase LCZ composites [4]. The material has high relative density, small grain size, higher thermal stability, and lower thermal conductivity, with fracture toughness values in the range of 2.13–2.5MPa·m$^{1/2}$. In addition, the Vickers hardness of LC40Z composites ranged from 8.68 ± 0.87 to 10.99 ± 0.23 GPa, and the fracture toughness ranged from 1.97 ± 0.15 to 2.4 ± 0.14 MPa·m$^{1/2}$. The denser microstructure had homogeneous grains and reduced porosity [5]. High-entropy rare earth niobates, the ReNbO$_4$/Re$_3$NbO$_7$ composite, was prepared via a solid-state reaction. The

high-entropy rare earth niobates exhibited excellent phase stability, higher fracture toughness and hardness, with a fracture toughness of $2.71 \pm 0.17$ MPa·m$^{1/2}$ and hardness of $9.46 \pm 0.24$ GPa, respectively. The high-entropy niobates exhibited high coefficients of thermal expansion, which were close to 7YSZ [6]. The quasi-binary $GdNbO_4/Gd_3NbO_7$ composites due to residual stress-activated ferroelastic domain switching and the fracture toughness were significantly improved with a toughness value of 2.76 MPa·m$^{1/2}$, which is currently the best for this series of high-entropy rare earth niobates [7]. These composites have low thermal conductivity, excellent mechanical properties, an appropriate coefficient of thermal expansion, and a comprehensive balance of properties in all aspects. The low thermal conductivity of the material, with a comprehensive balance of properties, provides an option for the design and manufacture of TBC materials and has excellent application potential.

The ceramic of the ZYTO system is a popular candidate for the next generation of thermal barrier coatings, with advantages such as thermodynamic stability [8,9] at high temperatures, resistance to harmful phase transformation, low thermal conductivity [10,11], and potential strength and toughness [12,13]. Therefore, the toughening mechanism of the ZYTO system has aroused widespread concern among scholars. As we know, ferroelastic toughening in the ZYTO system maximizes the toughness of ceramics and enhances their practicability in practical applications [14–17]. However, the fracture toughness of the ZYTO system is higher than some traditional ceramics but not compared with YSZ [18]; this weakness shortens its service life as a thermal barrier coating. Traditional 7-8YSZ's fracture toughness is about 4MPa·m$^{1/2}$ [19]. Recently, it was found that 7YSZ could reach a higher fracture toughness via a finely crystalline technique under sintering conditions at 1600 °C, the microstructure of which exhibits a composite phase composed of m-$ZrO_2$ and t-$ZrO_2$ [20,21]. In this paper, we focused on the mechanism that the second phase M-YTaO$_4$, affects toughening by changing the crack behavior; based on this, we attempted to determine the best toughening formula of M-YTaO$_4$ doping.

This paper studies the mechanical properties characterization and crack growth mode of the fluorite field and M-YTaO$_4$ field in the ZYTO system. The M-YTaO$_4$ solid solution was studied by XRD and EDS, the distribution of the second phase was observed under SEM, and several crack propagation behaviors were studied. The fine grain, phase stability, interface binding and residual stress of composites were shown to be influenced by the second-phase doping content. We tried to find some answers to several remaining problems. First, are the crack propagation behaviors and the fracture toughness sensitively dependent on the composition? Second are ferroelastic toughening and second phase toughening in cooperation or competition. Moreover, does the best toughening formula exist for this composite ceramic?

## 2. Material and Methods

ZYTO composite ceramic powders were prepared by chemical co-precipitation and calcination. First, five groups of YT1–YT5 samples were set in $x$ mol%Ta$_2$O$_5$ ($x = 10/20/30/40/50$) increments. All samples were prepared by a chemical to coprecipitate using precursor solutions of $ZrO(NO_3)_2$ ($\geq$99.99%), $Y(NO_3)_3$·6H$_2$O ($\geq$99.99%) and $TaCl_5$ ($\geq$99.99%). The pH of the solution was maintained above 10 to ensure the precipitation of all the mixed cations at the molecular level. The corresponding chemical equation for the development is as follows:

$$2ZrO(NO_3)_2 + 3Y(NO_3)_3 \cdot 6H_2O + TaCl_5 + 18NH_3 \cdot H_2O \rightarrow YTaO_4 + Zr_2Y_2O_7(\text{fluorite}) + 5NH_4Cl + 13NH_4NO_3 + 27H_2O$$

We separated the solution and hydroxide precipitate by centrifuge before cleaning twice with ethanol and drying in a drying oven for 12 h. To ensure the full conversion of tantalum-rich hydroxide to oxide, the pyrolysis was then conducted at 1300 °C for 5 h. The obtained powder was pressed into a single shaft at ~200 MPa and sintered in the air at 1500 °C for 10 h. The preparation process is shown in Figure 1.

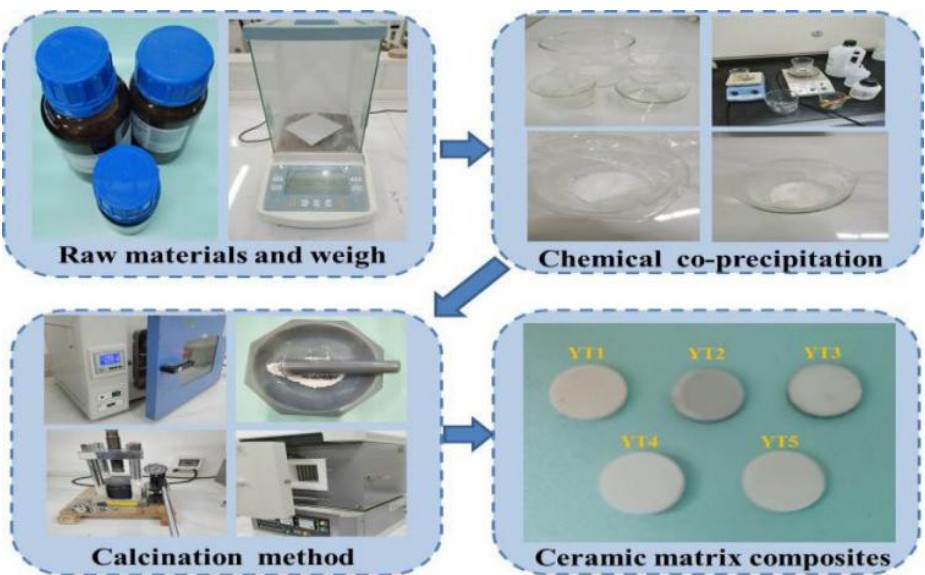

**Figure 1.** Flow chart of preparation process.

Vickers hardness ($H_V$) was measured by the indentation method. The indentation hardness test and analysis system (ZHVST-30F, Zhongyan, Shanghai, China) were used for measurement on the polished surface of the sample with a load of 10 s at 29.8 N. At least 6 effective indentations were made per sample. Fracture toughness ($K_{IC}$) was calculated by Ref. [22] based on the radial crack pattern of Vickers indentation:

$$K_{IC} = 0.16 H_V a^2 c^{-\frac{3}{2}} \tag{1}$$

where $H_V$ is Vickers hardness, $a$ is the half length of the diagonal of the indent, and $c$ is the half crack length measured from the middle of the dent to the crack tip, as shown in Figure 2. Figure 2 shows the secondary electron (SE, TESCAN MIRA3 LMH, Brno, Czech Republic) image, where the diamond indentation shape and cracks can be clearly seen.

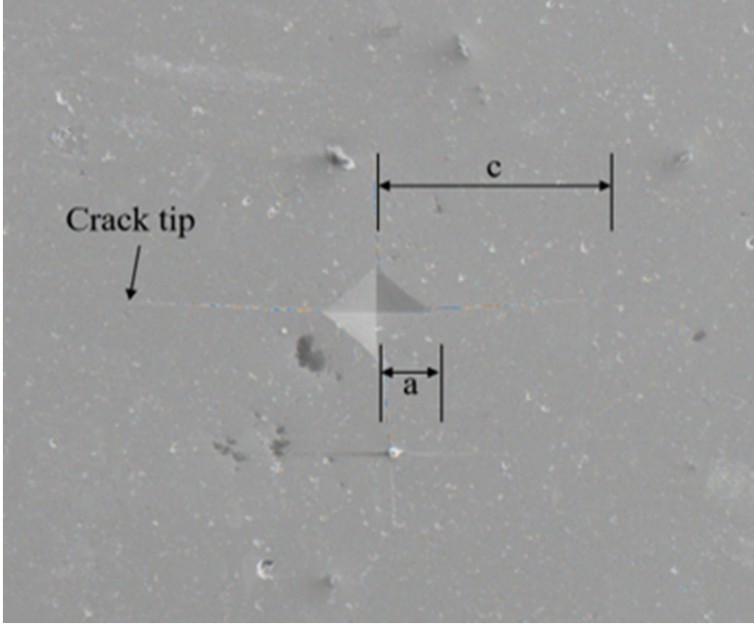

**Figure 2.** Secondary electron diagram of composite ceramic materials.

The critical energy release rate $\Gamma$ (fracture energy) refers to the ability to resist fracture, which is the actual crack propagation parameter, as a dynamic evolutionary variable indicating the energy required to propagate a certain length of the crack. The $\Gamma$ is N·m/m$^2$ or J/m$^2$. Therefore, $\Gamma$ is also understood as the energy provided by the system with each unit area of the crack propagation or the force provided by the system with each unit length of crack propagation. The $\Gamma$ is also called the driving force or the fracture energy. The formula is:

$$\Gamma = 2\xi^2 P \frac{a^2}{c^3} \tag{2}$$

where the value of $\xi$ is 0.016 ($\pm$0.004) calibration constant, $P$ is the loading force, $a$ is the half length of the diagonal of the indent, and $c$ is the average crack length measured from the indentation center.

Young's modulus ($E$) can be measured by the UMS Advanced Ultrasonic Material Characterization System (UMS-100, Rohde Schwarz, Munich, Germany). Young's modulus can be obtained by the following formula [23,24]:

$$E = \frac{V_L^2 \rho (1 + v)(1 - 2v)}{1 - v} \tag{3}$$

$$v = \frac{1 - 2\left(\frac{V_T}{V_L}\right)^2}{2 - 2\left(\frac{V_T}{V_L}\right)^2} \tag{4}$$

where $V_T$ is the transverse sound speed, $V_L$ is longitudinal sound speeds, $v$ is Poisson's ratio and $\rho$ is the density measured by Archimedes' principle, as shown in Table 1.

**Table 1.** Density of sample.

| Sample | Density $\rho$ (g·cm$^{-3}$) |
| --- | --- |
| YT1 | 5.09 |
| YT2 | 5.93 |
| YT3 | 6.22 |
| YT4 | 6.68 |
| YT5 | 6.95 |

## 3. Results and Discussion

### 3.1. Microstructure Characterization

The X-ray Diffraction (XRD, Ultimate IV, RIGAKU, Tokyo, Japan) patterns of ZYTO composites are shown in Figure 3. The XRD reflections of the fluorite phase and M-YTaO$_4$ are fluorites and scheelite structures. Figure 2 shows that with the increase in M-YTaO$_4$ content, the scheelite peak increases, indicating an increase in the scheelite phase amount. Only the fluorite phase and the M-YTaO$_4$ phase were detected in the composite, revealing no chemical reaction between the fluorite phase and M-YTaO$_4$. We analyzed the phase distribution by means of scanning electron microscopy (SEM, TESCAN MIRA3 LMH, Brno, Czech Republic) equipped with an energy dispersion spectrometer (EDS, X MAX20, Oxford Instruments, Oxford, UK). Surface SEM images of hot-etched ZYTO pellets are shown in Figure 4. In the image of YT1-YT4, grains with light are the M-YTaO$_4$, dark is the fluorite phase, and clear contrasts are observed. With the increase in the doping content, fluorite grain sizes further decreased.

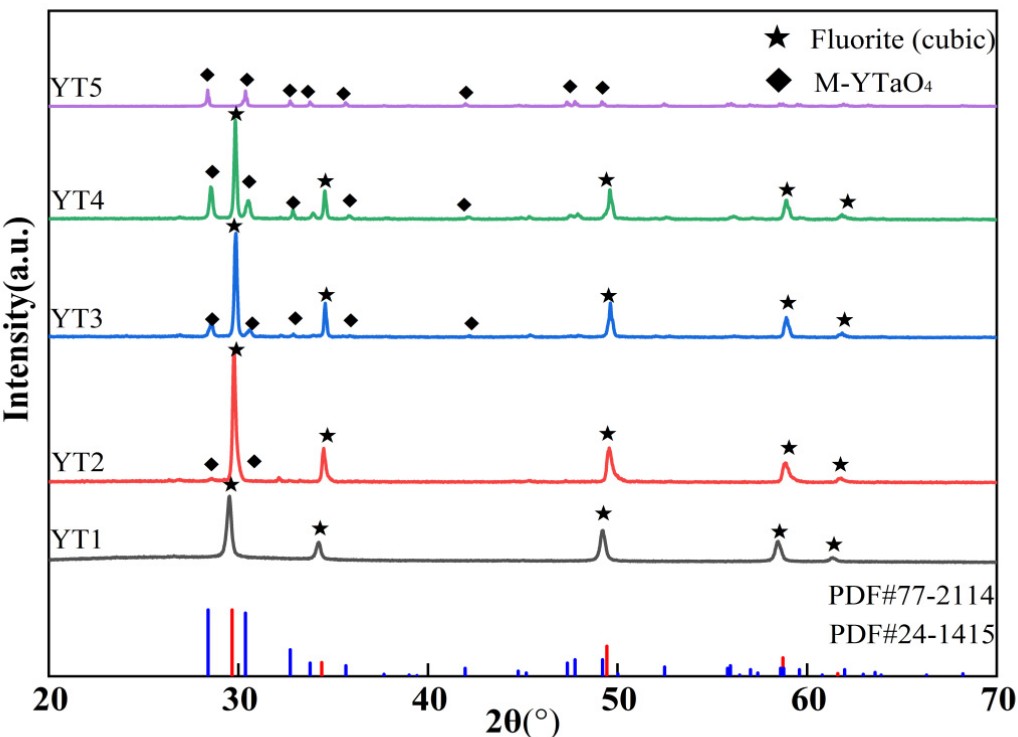

**Figure 3.** XRD pattern of fluorite phase and M-YTaO$_4$ phase composite at 1500 °C.

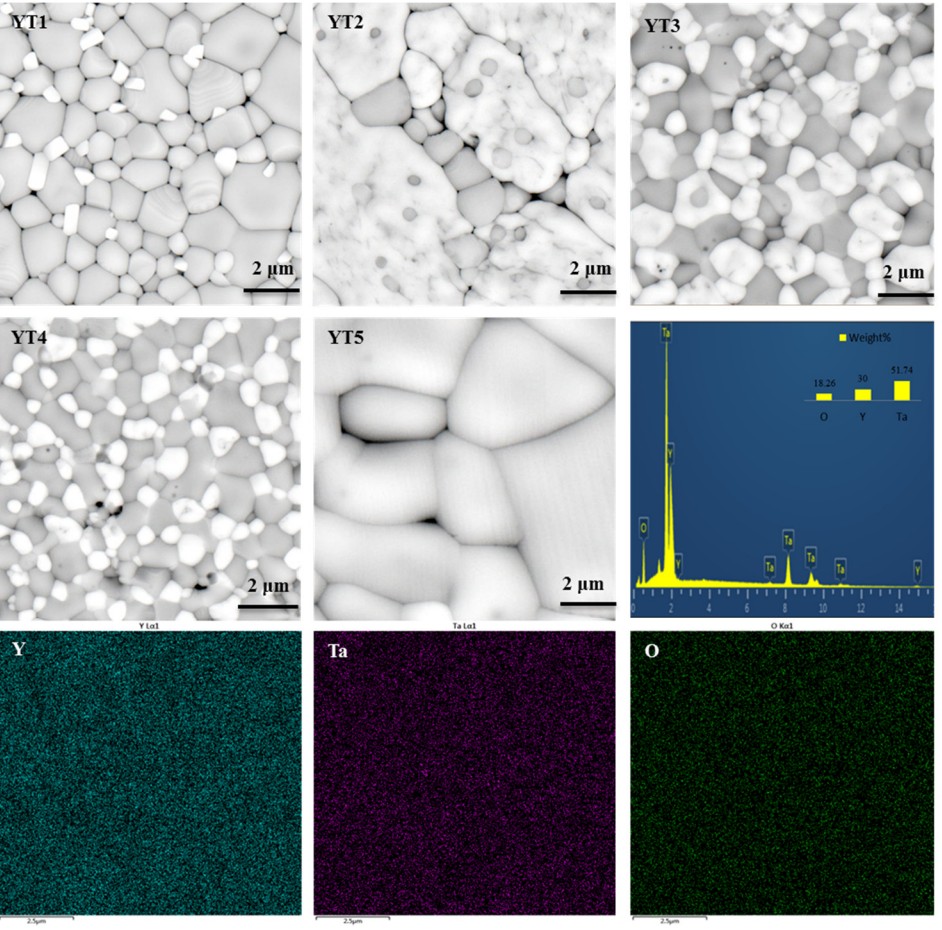

**Figure 4.** Surface BSE morphology images of Sample YT1-YT5 and EDS of the M-YTaO$_4$ grains in sample YT3.

The linear intercept method [25] was used to estimate the average grain size in the composite. As shown in Figure 5, the average size of fluorite grains showed a decreasing trend with an increase in the second phase. This indicated that the doping of M-YTaO$_4$ could inhibit the growth of fluorite grains during the sintering process. As shown in Figure 3, the interconnection of M-YTaO$_4$ grains occurred in YT3 and YT4, and the M-YTaO$_4$ grains in the composites became gradually larger. It can be noted in particular that when the doping amounts were YT3 and YT4, the fluorite and M-YTaO$_4$ grains in the composites were close in size. According to percolation theory, there is a critical range of the 14~16 vol% second phase volume fraction in the two-phase composites beyond which the second phase interconnection effect occurs [26,27]. In the present study, the M-YTaO$_4$ content was significantly higher than this range, which is why this phenomenon occurred. Furthermore, other studies have shown [28,29] that doping the second phase can refine the grain size of the matrix phase and enhance the mechanical properties of the composites.

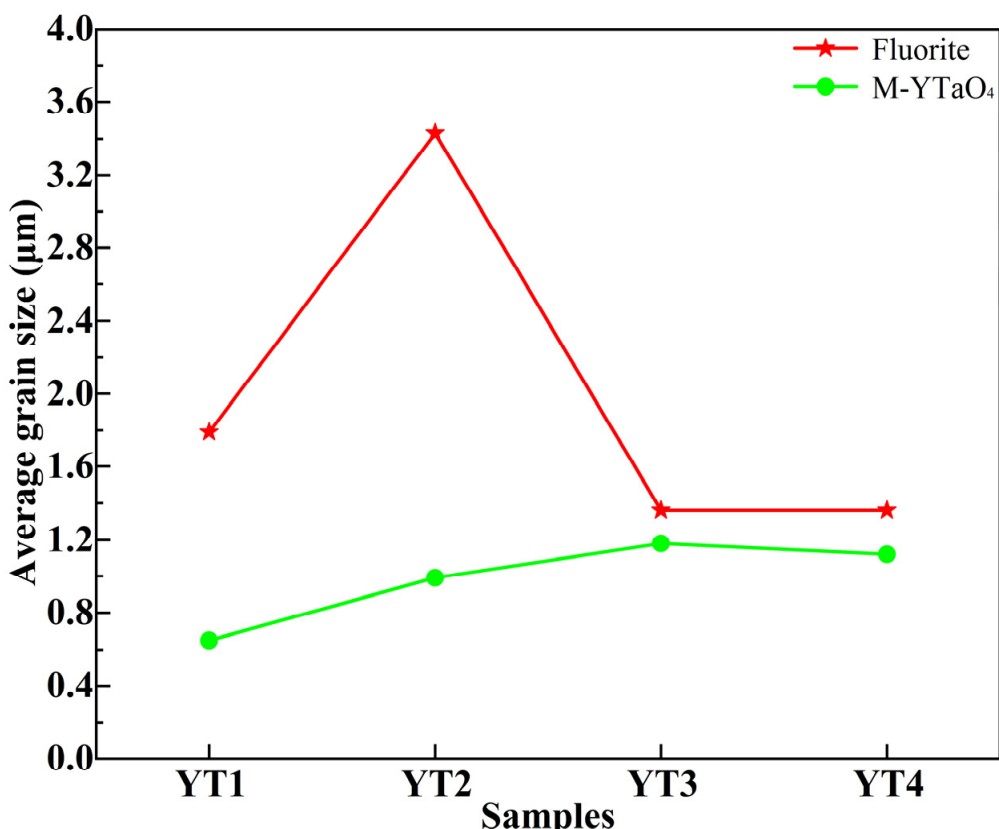

**Figure 5.** Comparison of the average size of M-YTaO$_4$ and fluorite grains in the composite.

### 3.2. Hardness and Fracture Toughness

Mechanical properties values obtained by the UMS Advanced Ultrasonic Material Characterization System and the indentation hardness test and analysis system are shown in Table 2. The Vickers hardness of ZYTO composites as a function of M-YTaO$_4$ doping is shown in Figure 6a. Obviously, the hardness decreased almost linearly with the doping of the second phase. The hardness decreased from 10.6 GPa to 4.5 GPa, which was higher than that of 8YSZ without additives (from 1.52 to 2.05 GPa) and was considerably lower than that of 8YSZ with additives CuO-TiO$_2$ (17.2 to 17.96 GPa) [30].



**Table 2.** Measured mechanical properties values of samples YT1–YT4.

| Sample | $V_L/m\cdot s^{-1}$ | $V_T/m\cdot s^{-1}$ | $E/GPa$ | $B/GPa$ | $G/GPa$ | $\upsilon$ | $H_V/GPa$ |
|--------|---------|---------|---------|---------|---------|------|---------|
| YT1 | 6095.24 | 3413.33 | 150.81 | 110.03 | 59.3027 | 0.2716 | 5.10 |
| YT2 | 6175.44 | 3450.98 | 179.80 | 131.98 | 70.6219 | 0.2730 | 8.31 |
| YT3 | 5803.92 | 3067.36 | 152.89 | 131.49 | 58.5221 | 0.3062 | 7.87 |
| YT4 | 5333.33 | 2580.65 | 119.86 | 130.69 | 44.4872 | 0.3471 | 5.83 |

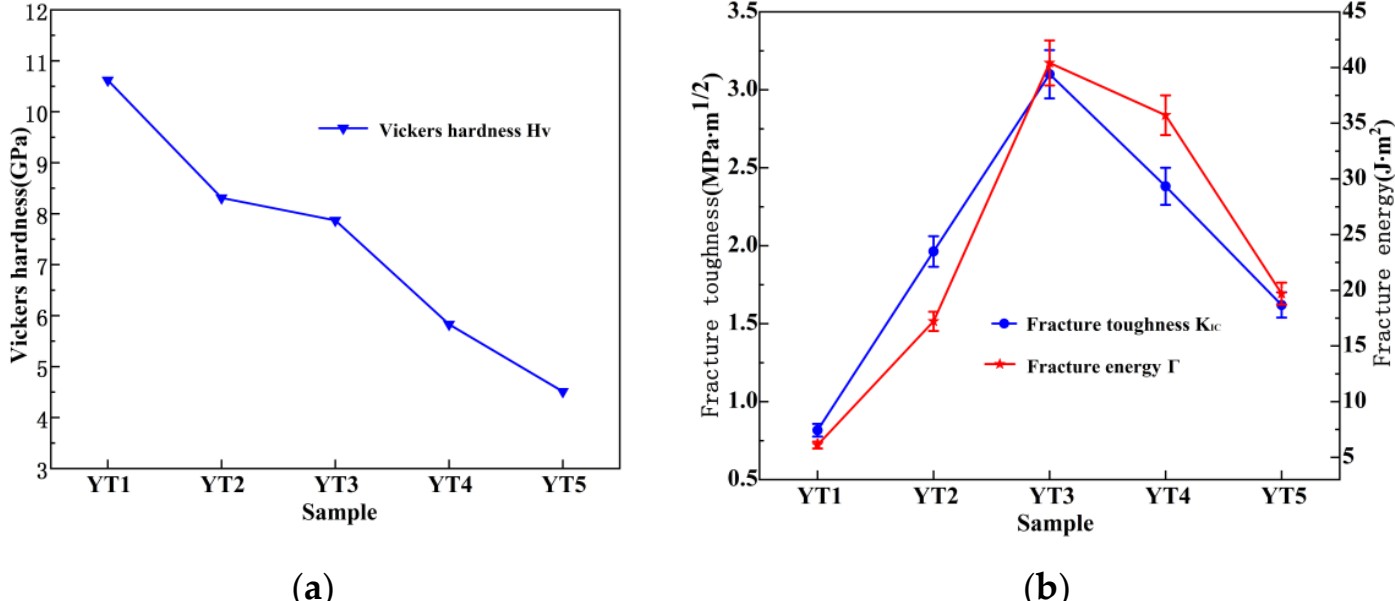

**Figure 6.** The relationship of Vickers hardness (**a**), fracture toughness and fracture energy (**b**).

Furthermore, due to the low density of the sample and more pores under the same load, the surface of the material was more prone to deformation, resulting in a sharp decline in hardness. Secondly, the porosity and cracks in the sample increased, and the brittleness of the material increased, which also affected the mechanical properties of the material. In addition, pores were apparent in ceramics, which could decrease the hardness of sintered specimens. As shown in Figure 3, a small amount of M-YTaO$_4$ doping was able to enhance the fluorite phase grain boundaries. The fluorite phase ceramic material improved its resistance to deformation in local areas. As a result, the hardness of YT1 was higher. As the amount of M-YTaO$_4$ doping increased, the hardness of the composites YT2-YT4 decreased instead compared to their fluorite phase counterparts. It could, thus, be inferred from the mixing law that the overall hardness approached that of M-YTaO$_4$ phase ceramics as the M-YTaO$_4$ doping content increased.

The trends in fracture toughness and fracture energy calculated by Equations (1) and (2) are shown in Figure 6b. The fracture toughness increased when increasing the M-YTaO$_4$ content and reached the highest value (approximately 3.1 MPa·m$^{1/2}$) with the samples YT3, which was almost 300% higher than that of the fluorite ceramic. At this moment, the fracture energy also reached a peak, and the indentation toughness was approximately equal to 42 J/m$^2$, which was comparable to 7-8YSZ's 45 ± 5 J/m$^2$ [31]. Fracture toughness and fracture energy followed a similar trend with increased doping, both increasing and then decreasing. The original increase in fracture toughness could be attributed to the grain size effect [32].

*3.3. Mechanism of Toughening*

Based on the above analytical results, we can conclude that the introduction of the second phase M-YTaO$_4$ into fluorite phase ceramics was beneficial in improving the hardness

and fracture toughness, although this change was not linearly proportional to the doping concentration. Further analysis of the toughening mechanism of composite ceramics revealed that the increase in fracture toughness was strongly related to residual stress, the interface state, crack propagation mechanism and ferroelastic switching.

### 3.3.1. Effects of the Residual Stress

In ZYTO composite ceramics, some studies have shown that the CTE (thermal expansion coefficient) of the second phase was higher than that of the matrix. When cooled from sintering temperature to room temperature, the base phase and the second phase produced compressive and tensile stresses, respectively, which is an average (effective) stress level. The displacement of the diffraction peak in the XRD pattern could reflect residual stress. Therefore, the peak displacement can exhibit whether compressive stress or tensile stress is generated. As shown in Figure 7a, the diffraction peaks of the second phase and the matrix moved in an opposite trend with the increase in the doping amount. For example, the M-YTaO$_4$ peaks moved at low angles to produce tensile stresses, while the matrix fluorite phase peaks moved at high angles to produce compressive stresses. This confirmed the formation of tensile stresses in the second phase and compressive stresses in the substrate. The actual situation at different grains and interfaces may be completely different due to the grain size, grain shape in terms of sharp notches, temperature gradient during cooling at the corresponding location, presence of stress micro-concentrators in the form of voids, micro-cracks, etc.

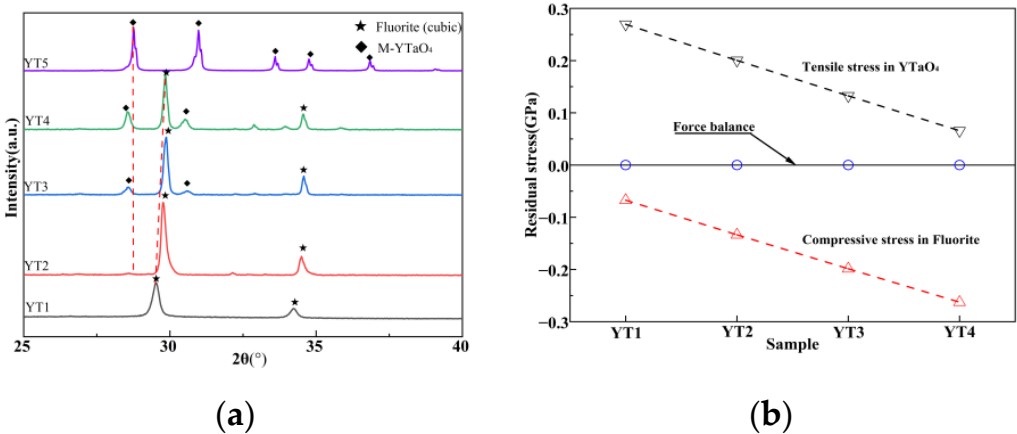

**(a)**                                              **(b)**

**Figure 7.** Slow-scan XRD pattern (**a**) of the composite at 2θ = 25°–40° as well as residual stress (**b**).

According to the model proposed by Taya [33], the residual stress in the composite can be calculated by the following equation:

$$\frac{\sigma_s}{E_m} = \frac{-2(1-x)\beta(\alpha_s - \alpha_m)(T_0 - T)}{(1-x)(\beta+2)(1+v_m) + 3\beta x(1-v_m)} \tag{5}$$

$$\frac{\sigma_m}{E_m} = \frac{2x\beta(\alpha_s - \alpha_m)(T_0 - T)}{(1-x)(\beta+2)(1+v_m) + 3\beta x(1-v_m)} \tag{6}$$

$$\beta = \frac{1+v_m}{1-2v_s}\frac{E_S}{E_m} \tag{7}$$

where the subscripts s and m represent the second phase and the matrix, respectively, *x* is the content of the second phase with Young's modulus *E*, Poisson's ratio *v*, thermal expansion coefficient *α*, tensile stresses *σs* and compressive stresses *σm*. $T_0$ and *T* represent the room temperature and sintering temperature, respectively. Table 3 lists the values of these parameters for calculating residual stress. The TEC of the fluorite phase is $(9.6 \times 10^{-6}\ \mathrm{K}^{-1})$, which is lower than that of the M-YTaO$_4$ phase $(10.7 \times 10^{-6}\ \mathrm{K}^{-1})$ [34].

Young's modulus and Poisson's ratio in the M-YTaO$_4$ phase were (177 GPa and 0.35), and that of the fluorite phase was (210 GPa and 0.3) [35]. Therefore, the calculated residual stresses are shown in Figure 7b. It is obvious that in the composite, the second phase grain was subjected to tensile stress, while the matrix was under compressive stress. With the increase in the second phase content, the tensile stress of the second phase grain decreased, while the compressive stress in the matrix increased. As we know, residual stresses during cooling can heal cracks and improve toughness [36].

**Table 3.** Calculation parameters of residual stress.

| Phase | E (GPa) | $\nu$ | A (K$^{-1}$) |
|---|---|---|---|
| Fluorite | 210 | 0.30 | $9.6 \times 10^{-6}$ |
| M-YTaO$_4$ | 177 | 0.35 | $10.7 \times 10^{-6}$ |

The synergistic influence of residual stress on crack growth behavior is schematically plotted in Figure 8. When the tensile stress was perpendicular to the direction of the crack extension, and the compressive stress was opposite to the direction of the tensile stress, the crack passed through the second phase, which consumed fracture energy and improved the toughness of the material (mode I). When the tensile stress direction was parallel to the interface, the crack was likely to propagate in the interface plane rather than along the initial path, resulting in crack deflection (Mode II). If the tensile stress was perpendicular to the interface, a new crack originating at the interface or crack bridging (mode III) may have occurred. For composite ceramics with different M-YTaO$_4$ content, their second phase distribution and residual stress distribution are different.

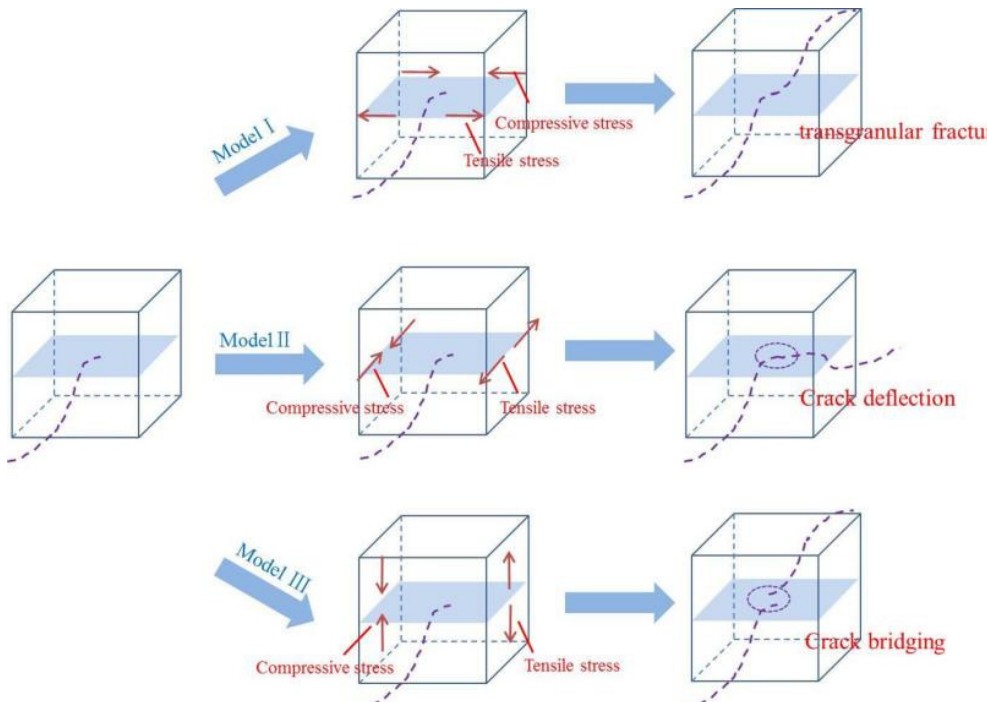

**Figure 8.** Schematic diagram of crack growth behavior under the synergistic action of tension and compression.

### 3.3.2. Effect of Interface Binding

Figure 9a shows how the fluorite phase exhibited a transgranular fracture mode, indicating weaker fracture toughness in the fluorite phase. Figure 9b shows that the composites exhibited a crack deflection mode, indicating stronger fracture toughness in the

M-YTaO$_4$ phase. When the crack propagated to position P, path 1 appeared along the interface between M-YTaO$_4$ and M-YTaO$_4$ grains and path 2 along the M-YTaO$_4$/fluorite grain boundaries. The existence of path 2 suggests that the interface strength of M-YTaO$_4$/fluorite was weaker than that of M-YTaO$_4$/M-YTaO$_4$.

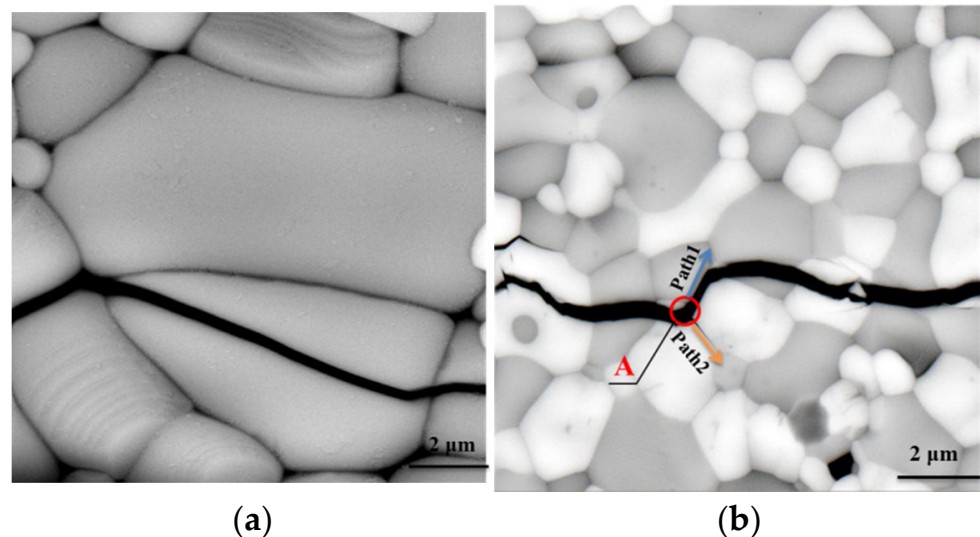

**(a)** **(b)**

**Figure 9.** Cracks propagation behavior in YT1 showing a transgranular fracture mode (**a**), and in YT2-YT4 showing an deflection mode triggering mechanism (**b**).

As described above, there were numerous interfaces between M-YTaO$_4$ and Fluorite grains in the composites. Therefore, the first problem was how does the M-YTaO$_4$/fluorite interface strength affect the fracture toughness of the composite? Unfortunately, measuring the interface strength between the grains is almost impossible. In order to prove that the change in doping had an effect on Young's modulus of the composite, a linear analysis was carried out following Voigt's work [37]. Voigt provided a qualitatively analyzed model to investigate how the interface strength affected Young's modulus. The equation for the related calculation is as follows:

$$E_{\text{cal}} = xE_s + (1-x)E_m \tag{8}$$

where $s$ and $x$ represent the content of the second phase and the second phase, respectively, and m is the matrix phase. Table 4 shows Young's modulus of the composites calculated values and the experimental values.

**Table 4.** Young's modulus of sample.

| Sample | Young's Modulus $E$ (GPa) | |
|---|---|---|
| | **Calculated Values** | **Measured Values** |
| YT1 | 203.0 | 150.8 |
| YT2 | 196.8 | 179.8 |
| YT3 | 190.2 | 152.9 |
| YT4 | 183.6 | 119.9 |
| YT5 | 177.0 | 78.3 [*] |

[*] Value obtained from the three points loading flexure.

Obviously, Young's modulus is measured by a value lower than the calculated value. However, Voigt believed that strengthening interface bonding meant increasing Young's modulus [26]. Figure 10 shows that the measured value first increased and then decreased, reaching a maximum value at YT2. Combined with Figure 4, the nonlinearity behavior of Young's modulus may have been caused by M-YTaO$_4$ grains, which were wrapped by

fluorite phase grains at YT2. The fluorite phase grains and M-YTaO$_4$ grains were closely combined to enhance the interface's strength. However, as the second phase increased, the second phase grain became larger, and the pores increased. This led to the rapid decline of Young's modulus. In addition, both the calculated and measured values of Young's modulus showed a downward trend after adding M-YTaO$_4$. As we know, a low Young's modulus can result in higher strain tolerance in the thermal barrier coating, effectively alleviating the stress caused by thermal shock [38]. Based on the above discussion, we could conclude that M-YTaO$_4$ doping had a significant impact on the interface bonding of composite materials.

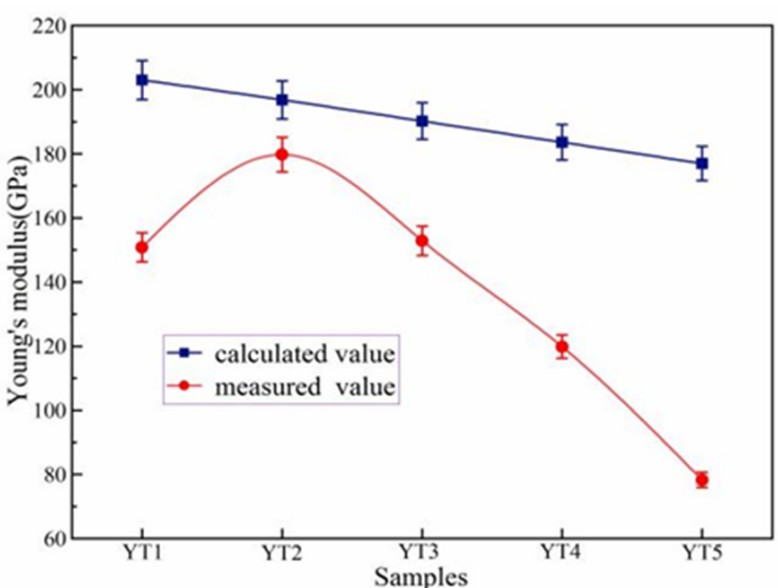

**Figure 10.** The relationship between the measured value of Young's modulus and the value calculated according to the Voigt model.

### 3.3.3. Analysis of Crack Propagation Mechanism

In this study, composites YT1 mainly revealed a transgranular fracture mode, as shown in Figure 11a. Cracks were initiated in the matrix and crossed through the fluorite phase grain. The crack extension of the composites YT2-YT4 is shown in Figure 11b–d. When the crack encountered the second phase (M-YTaO$_4$ grains), crack deflection, bridging and bifurcation were the main modes of propagation. The crack needed to consume more energy for propagation because of the higher interface bonding of M-YTaO$_4$ grains, giving rise to an enhanced toughness, which could account for the obviously high toughness of the composite. Thus, cracks deflection, bridging and bifurcation were considered important toughening mechanisms. Hence, introducing the second phase was a primary contributor to the improved toughness of the ceramic materials.

Due to the unfixed angle between the tensile interface and the layer interface, when the crack encountered the M-YTaO$_4$ grain interface, it could penetrate or deflect, adding new free surfaces and releasing more fracture energy. Figure 11 exhibits three fracture modes, crack deflection, intergranular fracture, and crack bridging, indicating that the strength of M-YTaO$_4$ grains was larger than those of the fluorite phase, which led to the toughness of the two-phase region is improved. By analyzing the crack propagation behavior in the composite material, the bonding strength of the two-phase interface could be recognized. Similar phenomena have been observed by other researchers [39].

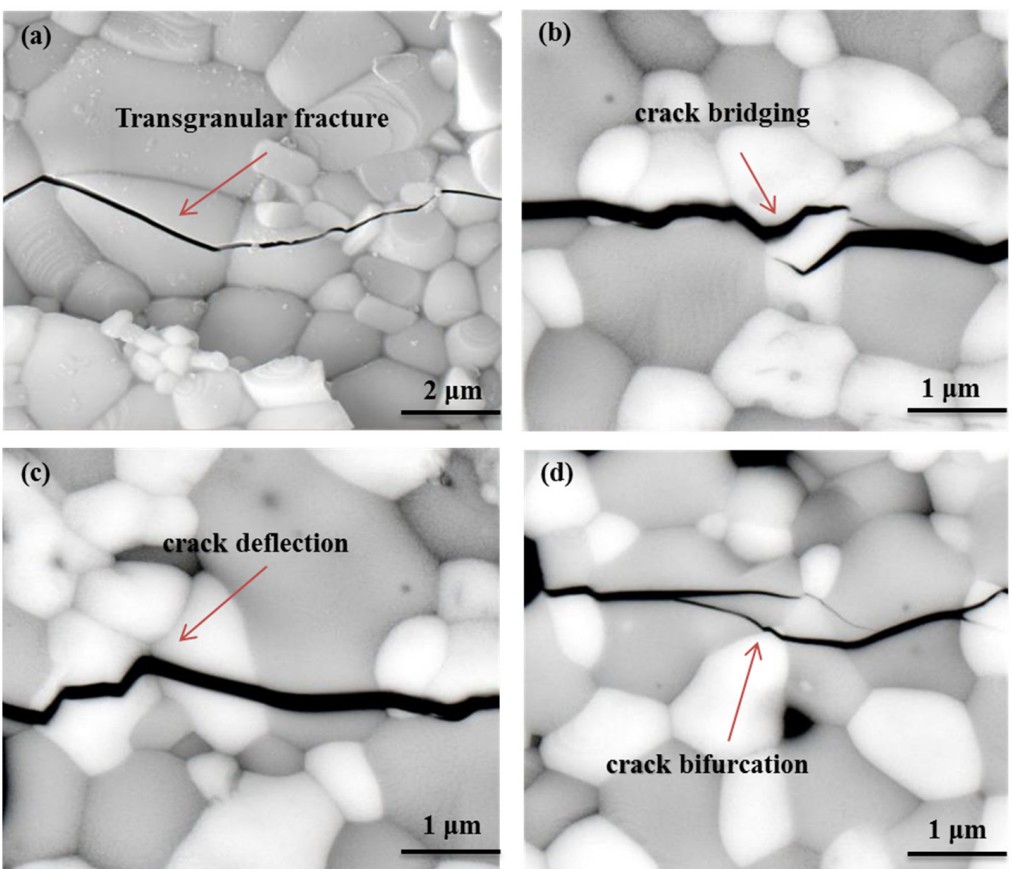

**Figure 11.** Transgranular fracture in YT1 (**a**), crack bridging (**b**), deflection (**c**), and bifurcation (**d**) in YT2-YT4 due to second phase doping.

3.3.4. Toughening Mechanism of Ferroelastic Domain of YTaO$_4$

M-YTaO$_4$ is a stable ferroelastic structure at room temperature. Due to the T → M phase transition of YTaO$_4$ being ferroelastic, it is of a continuous second-order nature [40]. Previous research states that the T → M phase transition of YTaO$_4$ is ferroelastic, the high symmetry phase (T) breaks into the low symmetry phase (M) at the Curie point, and the corresponding spontaneous strain is:

$$e(S_1) = \begin{pmatrix} e_1 & e_6 & 0 \\ e_6 & e_2 & 0 \\ 0 & 0 & e_3 \end{pmatrix} \tag{9}$$

where:

$e_1 = \frac{c_M \sin\beta}{a_T} - 1; e_2 = \frac{a_M}{a_T} - 1; e_3 = \frac{b_M}{c_T} - 1; e_6 = -\frac{c_M \cos\beta}{2a_T}$.

There are two different orientation states in the monoclinic phase related to the two ferroelastic variants [41,42]. We observed that variants M1 and M2 could switch to each other in the process of crack propagation. The crystallography of the ferroelastic transition and the ferroelastic switching is present in Figure 12, the deformed geometry of M1 was obtained by a tetragonal inclusion undergoing an eigenstrain of e(S$_1$), and the M2 variant was obtained in the same way, and was associated with the eigenstrain of e(S$_2$). Alternatively, one could rotate the M1 around the $c_T$ axis 90° clockwise to obtain M2, which is termed ferroelastic switching. To corroborate the above theory, the XRD analysis of the lattice parameters (a b c), volume (*V*) of YTaO$_4$, and Fluorite phase were obtained, as shown in Table 5.

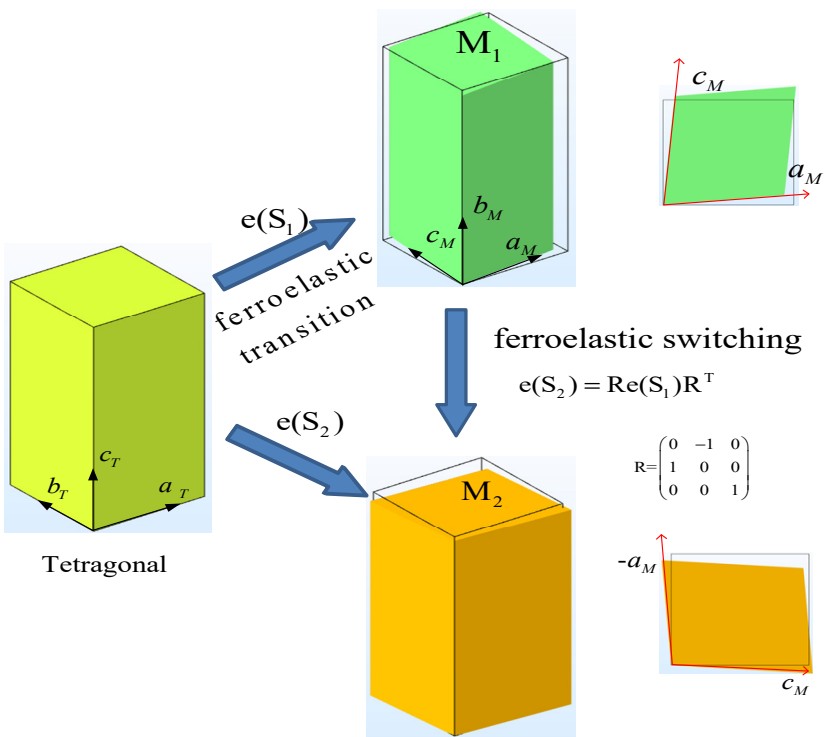

**Figure 12.** Schematic diagram of ferroelastic transition (T → M) and switching (M1–M2) of YTaO$_4$. The deformed geometries of M1 and M2 were reconstructed using a finite element method. The lattice constants of tetagonal crystal are borrowed from an ab inito calculation [43], and the eigenstrain of Equation (9), associated with a deformed inclusion, (M1) could be derived from the lattice constants of YT5 of Table 5.

**Table 5.** Calculated lattice parameters (a b c) and volume (*V*) of YTaO$_4$ and Fluorite phase.

| Sample | M-YTaO$_4$ | | | | Fluorite | |
|--------|--------|--------|--------|-------------|----------|-------------|
| | a (Å) | b (Å) | c (Å) | vol (Å$^3$) | abc (Å) | vol (Å$^3$) |
| YT1 | 5.326 | 10.932 | 5.05 | 292.65 | 5.2168 | 141.98 |
| YT2 | 5.3189 | 10.9038 | 5.0666 | 292.59 | 5.2098 | 141.41 |
| YT3 | 5.3324 | 10.9317 | 5.0423 | 292.59 | 5.2106 | 141.46 |
| YT4 | 5.3334 | 10.9337 | 5.0389 | 292.5 | 5.2099 | 141.42 |
| YT5 | 5.3252 | 10.9313 | 5.0538 | 292.83 | - | |

Figure 13a,b shows that the ferroelastic domain existed of parallel striped structures in the pure YTaO$_4$ ceramic (samples YT5) and also indicated that there was always only one polarization direction in one grain when the grain was far from the crack. Furthermore, we found that the ferroelastic structure became more pronounced as the grains became larger. Figure 13c shows that M1 and M2 coexisted in one grain and were nearly perpendicular to each other, which agreed with theoretical studies. Therefore, the M2 variant transformed into the M1 variant and could be seen as a sort of rotating twinning process associated with the rotation described in Figure 12. Ferroelastic domains with the different polarization direction (A and B) were observed near the crack tip in the bridging process, as shown in Figure 13c, which indicated that the factors to trigger ferroelastic switching were not only dependent on the crack issue, but the grain size and grain boundary geometry may have also affected the switching. Ferroelastic toughening was evidenced in Figure 13d when the crack penetrated the grain, and two polarization directions (C and D) were observed.

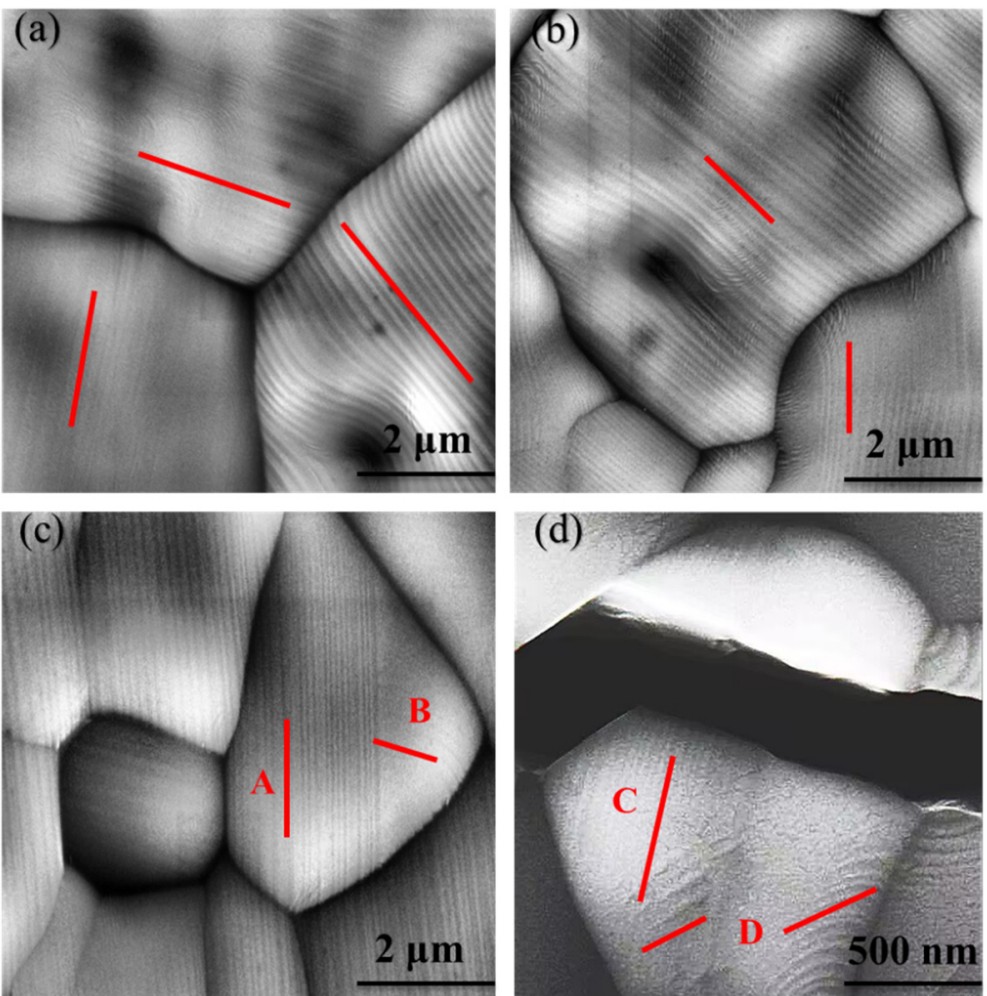

**Figure 13.** The sample YT5 pure M-YtaO$_4$ ferroelastic domain structure was observed by SEM (**a**,**b**), the twin grain visible in the figure is the M phase resulting from the displacement phase transformation of t-YTaO$_4$ during cooling. Ferroelastic switching in the bridging part of the crack tip in pure YTaO$_4$ (**c**). The sample YT3 is subjected to external stress and the ferroelastic domain structure is transformed (**d**).

As we know, ferroelastic switching can be seen as a sort of mechanical twinning [44]. Twinning stress was influenced by grain size, followed by the Hall–Petch law [45]. An increase in grain size served to decrease the critical twinning stress. In the experiments, we indeed found that, in the larger grain, ferroelastic switching near the crack was more deterministic than in the small grains. Therefore, we believe that ferroelastic toughening occupies an important position compared to others which can affect the toughness of the composite ceramic with an increase in m-YTaO$_4$ doping.

## 4. Conclusions

In this study, ZYTO composite ceramic materials were prepared by the chemical coprecipitation method. The microstructure of ZYTO was studied, as well as the mechanical properties and toughening mechanisms. The following conclusions can be drawn:

ZYTO composite ceramics show excellent phase stability and no chemical reactions between the two phases. The grain size of the two phases was gradually consistent with increasing the M-YTaO$_4$ doping concentration. The M-YTaO$_4$ phase refined the fluorite phase grain.

Through calculation and analysis, both the density and porosity of the ZYTO composite ceramics increased with increasing doping. The hardness decreased almost linearly with

the doping of M-YTaO$_4$, from 10.6 to 4.5 GPa. The fracture toughness first increased and then decreased with the M-YTaO$_4$ doping concentration, and YT3 possessed the highest fracture toughness of 3.1 MPa·m$^{1/2}$.

The residual stress, interface state, crack propagation mechanism and ferroelastic switching had important effects on the toughness of ZYTO composite ceramics. In this case, due to the introduction of M-YTaO$_4$ grains, the toughness of ZYTO composite ceramics was impacted by second-phase toughening and ferroelastic toughening, which helped to further improve the fracture toughness. We believe that the second phase of toughening and ferroelastic toughening played a dominant role in improving the toughness of the material.

**Author Contributions:** Writing—original draft, X.F.; Writing—review and editing, F.Z., W.Z. and Z.P. All authors have read and agreed to the published version of the manuscript."

**Funding:** This research is sponsored by National Natural Science Foundation of China (NSFC) under Grant No. 52171015, and Science and Technology Innovation Program of Hunan Province (Grant No. 2022RC1082).

**Institutional Review Board Statement:** Not applicable.

**Informed Consent Statement:** Not applicable.

**Data Availability Statement:** Not applicable.

**Conflicts of Interest:** The authors declare no conflict of interest.

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
