# Peer review of "Mechanical Properties and Toughening Mechanisms of Promising Zr-Y-Ta-O Composite Ceramics"

_coatings, doi:10.3390/coatings13050855_

Round 1

Reviewer 1 Report

The authors deal with the topical problem of crack resistance and mechanical strength of oxide multicomponent coatings, which are of interest both in the aerospace industry and in other gas turbine technologies. Indeed, modern materials often require functional coatings that can provide increased heat resistance and at the same time have a minimum thickness. I believe that the manuscript is prepared with high quality, and the material is presented in a scientific language, logically and clearly. Nevertheless, I think the article will look even more solid if the authors of the proposed diffraction patterns give a table of values for the lattice constant, unit cell volume, as well as refinements by the Williamson-Hall method with the calculation of the parameters of the coherent scattering region and residual stresses. This is an important element that would organically fit into the current work.

Important note! From the presented SEM images, it is not obvious where the grains of one phase are, and where the other. I could never tell them apart by contrast. Please provide explanations on the pictures.

I believe that such methods as polarizing microscopy together with EDX analysis at grain interfaces would allow to deepen the understanding of the mechanisms of cracking. It is known that the stoichiometry can change significantly at the grain boundaries. I do not insist on these measurements, but they would greatly strengthen the ground of your judgments.

In the text of the manuscript on line 60, the first letter in the new sentence should be corrected. Also in table 2 there is a hieroglyph, the meaning of which is not obvious to readers.

Reviewer 2 Report

The authors have studied mechanical properties, microstructural features, and toughening mechanisms of Zr-Y-Ta-O composite ceramic which is considered as the candidate for next generation thermal barrier coatings. The research is well designed but presented not very clearly. A quite short comparative analysis of existing publications concerning the tasks set in the work is performed. The methodological section of the manuscript is presented in sufficient detail but some issues should be explained. The authors used the modern equipment for test of samples as well as visualization and assistance in the interpretation of the obtained results. They found that M-YTaO4 refines the fluorite phase grain and strengthens the grain interface in the composite ceramic. Besides, the ferroelastic toughening mechanism as one of the competing toughening mechanisms in the composite was revealed. Along with this mechanism, other features of composite fracture were indicated, namely cracks deflection, bridging, and bifurcation, promoting an increase in fracture toughness of the composite.

However, some shortcomings should be corrected to make the manuscript acceptable for publication in Coatings.

(1) Most of the figure captions should be corrected. They should not contain the word “show/shows” in the beginning, should start with a capital letter and be independent phrases. The caption to Figure 1 does not explain what the figure means. Please fix this.

(2) The authors provided Ref.[12] from where they took the formula (1) for calculating fracture toughness of ceramics. However, in Ref.[12], there only is a reference to an article by Evans and Charles (1976) https://doi.org/10.1111/j.1151-2916.1976.tb10991.x. Therefore, it is expected that the authors substantiate the way they derived the formula (1) of such unusual form using an approach given by Evans and Charles (1976).

(3) Lines 3234: In the sentence “However, the fracture toughness of ZYTO system is higher than some traditional ceramics but not compared with the YSZ [7, add References], this weakness will shorten its service life as thermal barrier coating.” the authors should add more new references for substantiation of this advantage of YSZ ceramics. They can use the following articles in which a comparatively wide range of sintering temperatures for YSZ ceramics is analyzed in terms of the optimization of microstructure and mechanical properties: https://doi.org/10.3390/ma15082707, https://doi.org/10.1016/j.matdes.2023.111908. This will increase the weight and significance of the research.

(4) Table 1, the column “Density”: taking into account the value range, it seems that the unit (g·m-3) is wrong. It should be (g·cm-3). Please fix this.

(5) Table 2: If the authors provide units for corresponding parameters in the table caption, they should unify the style either by adding or removing units to the parameters in the upper row of the table itself. The column “VT”: The character other than the English alphabet should be removed. For the ν parameter, it is enough to provide only four decimals. For E and G parameters, it is enough to provide only two decimals. Besides, a column “Sample” (or “Ceramics”) should be added as the first column containing YT1-YT4 marking.

(6) Figure 7: In the figure body, the phrase “Crack bridgin” should be corrected. It should be “Crack bridging”. Please fix this.

(7) Figure 8(a): A crack path across a big grain can be clearly seen, whereas in the figure caption it is written that “Cracks propagation behavior in fluorite showing an intergranular fracture mode (a)”. This discrepancy should be fixed or explained clearly.

(8) Figure 11: In the figure caption, the authors should provide an explanation of what the text “M1”, “M2”, and arrows indicate.

(9) Figure 12: In the figure caption, the authors should provide an explanation of what the text “A”, “B”, “C”, “D”, and arrows indicate.

In my opinion, English language of this manuscript should be significantly improved.

Reviewer 3 Report

The manuscript has proper potential for accepting in this journal after minor revision. The questions, issues and recommendations are as follows.

1.       In the introduction section the TBCs be completely introduced.

2.       Also, in the introduction section, different research work, which used various TBC and substrates be introduced.

3.       The mentioned toughening mechanisms in the introduction section like phase transition toughening and etc. be completely introduced.

4.       It is recommended that a schematic figure for showing the chemical structures, reactions and fabrication process be added in the section 2.

5.       Some new references, which have been published in 2022 and 2023 be added in the manuscript. 

1.       The authors used the active sentence in line 47. Be modified.

2.       Similar previous issue, in line 60, the authors used the active sentence. Be checked whole manuscript and used the passive sentences.

Round 2

Reviewer 2 Report

All the reviewer’s comments were taken into account by the authors. The manuscript can now be accepted for publication in Coatings.

Minor editing of English language required.